# Dried Ginger Extract Restores the T Helper Type 1/T Helper Type 2 Balance and Antibody Production in Cyclophosphamide-Induced Immunocompromised Mice after Flu Vaccination

**DOI:** 10.3390/nu14091984

**Published:** 2022-05-09

**Authors:** Jihyun Kim, Hoyoung Lee, Sooseong You

**Affiliations:** KM Convergence Research Division, Korea Institute of Oriental Medicine, Daejeon 34054, Korea; kimjihyun763@naver.com (J.K.); lhoyoung@nate.com (H.L.)

**Keywords:** influenza vaccine, dried ginger, cyclophosphamide, antibody production, Th1/Th2

## Abstract

Dried ginger (*Zingiberis Processum Rhizoma* (ZR)) is frequently used to prevent or treat common cold and flu. This study aimed to investigate the influence of ZR extracts on influenza-specific antibody production in cyclophosphamide (Cy)-induced immunocompromised mice. Female BALB/c mice were injected three times with saline or Cy. To investigate the effect of ZR, either distilled water or ZR was administered orally to mice daily for 10 days after Cy injection. After ZR administration, the mice were immunized with the 2017/2018 influenza vaccine. Pretreatment with ZR extracts enhanced influenza-specific antibody production in Cy-induced immunocompromised mice after flu vaccination and restored the influenza antigen-specific T helper (Th) type 1/Th2 balance to the normal state. Further, ZR suppressed the eosinophil enrichment caused by Cy injection in the spleen. We demonstrated that ZR can be used to increase antibody production in immunocompromised individuals before vaccination.

## 1. Introduction

According to the World Health Organization, influenza causes 3–5 million severe cases and 290,000–650,000 influenza-related respiratory deaths worldwide annually [1]. In particular, immunocompromised individuals who undergo transplantation or chemotherapy are at a high risk of morbidity and mortality from influenza infection because of impaired host defense [2]. For this reason, many expert guidelines recommend annual influenza vaccination for immunocompromised individuals. Unfortunately, these individuals are still vulnerable to influenza-associated complications despite being vaccinated because of their impaired humoral vaccine responses [3]. To resolve this problem, new formulations, including adjuvanted vaccines and high-dose vaccines, have been developed to improve the effectiveness of influenza vaccines in immunocompromised individuals [4,5,6,7]. Although these attempts could enhance the immunogenicity of the influenza vaccine, weakened immune responses to other pathogens are still ongoing in these individuals. Therefore, the development of a method for restrengthening the immune system is needed to enhance the body’s ability to produce adequate antibodies after vaccination without an adjuvant or a high dose.

Traditionally, ginger is used to prevent or treat common cold and flu [8], but in recent times it has also been reported to exhibit therapeutic potential against COVID-19 through modulation of unbalanced T-cell responses [9]. Ginger has been shown to have antiviral, anti-inflammatory, and antioxidant properties [10,11,12,13]. It has been reported that its valuable functional ingredients like gingerols, shogaol, and paradols have anti-cancer potential [14,15]. Intraperitoneal injection of ginger extracts before an ovalbumin airway challenge suppressed T helper type 2 (Th2)-mediated immune responses to allergens in mice [16]. CD4^+^ T cell-mediated immune responses play an important role in the defense against influenza infection [17]. Previous studies have shown that the induction of the Th1 cell response to interferon-gamma (IFN-γ) enhances antibody production after vaccination [17,18]. If the ginger could potentiate the Th1 response by inhibiting the Th2 response, it could be a strategy for restrengthening the immune system to increase antibody production against the influenza vaccine in immunocompromised individuals.

Dried ginger (*Zingiberis Processum Rhizoma* (ZR)) contains various bioactive compounds, including gingerols and shogaols [19], and has medicinal properties that alleviate the symptoms of arthritis by regulating the genes encoding proinflammatory cytokines and chemokines [20]. Interestingly, dried ginger has higher antioxidant properties with different constituents than fresh ginger does because of the heating process [21]. Our study aimed to assess the effect of dried ginger extracts on flu-specific antibody production and Th1/Th2 responses in cyclophosphamide (Cy)-induced immunocompromised mice after flu vaccination.

## 2. Materials and Methods

### 2.1. Mice

All experiments and analyses were conducted in accordance with the relevant guidelines and regulations. Experimental animal protocols were approved by the Institutional Animal Care and Use Committee of the Korea Institute of Oriental Medicine, Daejeon, Korea (approval number 17-070). BALB/cAnN female mice (8 weeks old) were obtained from Oriental Bio Co. (Seongnam, Korea) and housed under specific pathogen-free conditions with freely available food and water.

### 2.2. Preparation of Dried Ginger Extracts

Dried ginger (ZR) was purchased from Omniherb (Yeongcheon, Korea) and extracted with 70% ethanol at 82 ± 2 °C for 3 h in an herb extractor (COSMOS-660, KyungSeo Machine Co., Incheon, Korea). The ethanol extracts were filtered and freeze-dried using a freeze dryer (yield = 11.65%) and analyzed by high-performance liquid chromatography to determine four constituents of ZR extracts, including zingerone, 6-gingerol, 8-gingerol, and 10-gingerol (Figure 1).

### 2.3. Immunization

The cytotoxic drug Cy (Sigma-Aldrich, St. Louis, MO, USA) was used to induce an immunocompromised state. Groups of eight mice were injected intraperitoneally three times every other day with saline (*n* = 8, non-treated group) or Cy (150 mg/kg). To investigate the effect of ZR, either distilled water (*n* = 8, Cy group) or 500 mg/kg ZR (*n* = 8, Cy + ZR group) was administered orally to mice daily for 10 days after Cy injection. On the day following the last oral administration of ZR, mice were immunized intramuscularly with inactivated Fluzone^®^ 2017/2018 influenza vaccine (Seoul, Korea) stocks containing four different strains (30 µg/mL: A/Michigan/45/2005(H1N1)pdm09-like strain, A/HongKong/4801/2014(H3N2)-like strain, B/Phuket/3073/2013, and B/Brisbane/60/2008-like stain). Serum and splenocytes were collected 4 weeks after immunization to measure influenza-specific antibody titers and perform flow cytometric analysis, respectively.

### 2.4. Measurement of Influenza-Specific Antibody Subclasses by Enzyme-Linked Immunosorbent Assay

Influenza-specific antibody responses were determined by enzyme-linked immunosorbent assay (ELISA) using the Fluzone vaccine as the coating antigen. Subsequently, 96-well ELISA plates (Nunc-Immuno Plate, Nunc Life Technologies, Basel, Switzerland) were coated with 1 µg/mL of Fluzone vaccine at 4 °C overnight. Serum samples were diluted (1:10,000) in Dulbecco’s Phosphate Buffered Saline with 5% bovine serum albumin. Briefly, horseradish peroxidase-conjugated goat anti-mouse IgG, IgG1, or IgG2a antibodies (Abcam, Cambridge, UK) were used as secondary antibodies to determine the total IgG or IgG isotype antibodies at 37 °C for an hour. The 3,3′-diaminobenzidine substrate (BD Biosciences, CA, USA) was used to develop color, and 2 M sulfuric acid was used to stop the color reaction. The optical density was measured at 450 nm using an ELISA reader.

### 2.5. Isolation and Culture of Splenocytes for Flu Antigen Stimulation

Mice were euthanized with isoflurane, and the spleen was removed and placed in Roswell Park Memorial Institute (RPMI) media (Gibco, Paisley, UK) with 1% penicillin/streptomycin (P/S; Lonza, GA, USA) on ice. Splenocytes were isolated using a 100 µm strainer, and red blood cells (RBCs) were lysed with RBC lysis buffer (BioLegend, CA, USA). Splenic cells (2 × 10^6^ cells/200 µL/well) were cultured in RPMI supplemented with 1% P/S, 10% fetal bovine serum, anti-CD 28 (1 µg/mL, BioLegend, San Diego, CA, USA), and the absence or presence of influenza antigen (1 µg/mL each, Fluzone vaccine) for 2 days using 96-well U bottom plates (Nunc Life Technologies). After 2 days, splenocyte supernatants were collected and stored at −70 °C.

### 2.6. Flow Cytometric Analyses of Isolated Splenic Cells

Isolated splenocytes (2 × 10^6^ cells) were resuspended in diluted zombie aqua solution (BioLegend) for live/dead discrimination. After anti-CD16/32 antibody (BioLegend) staining for Fc receptor blocking, splenocytes were stained with anti-CD3-FITC, anti-CD4-BV605, anti-CD8a-PerCP-Cy5.5, anti-CD19-APC, anti-CD11b-Percific Blue, anti-CD11c-PE-Cy7, anti-Ly6G-PE, and anti-Ly6C-APC-Cy7 (all from BioLegend). Stained cells were analyzed using a BD LSRFortessa Cell Analyzer (BD Biosciences, San Jose, CA, USA). We identified T cell (CD3+CD4+, CD3+CD8+), B cell (CD3-CD19+), dendritic cell (CD3-CD11c+), monocyte/macrophage (CD3-CD11b+Ly6C+), neutrophil (CD3-CD11b+Ly6G+), and eosinophil (CD3-CD11b+Ly6C-SSCHi) subsets in the mouse spleen, according to Ly6C/Ly6G-based strategy [22].

### 2.7. Cytometric Bead Array of Splenocyte Supernatants after Influenza Antigen Stimulation

The cytometric bead array (CBA) flex set (BD Biosciences) was used to quantify the levels of secreted cytokines in splenocyte supernatants according to the manufacturer’s protocol. Diluted supernatants (1:4) were prepared, and the following cytokines were measured: IFN-γ, tumor necrosis factor (TNF), interleukin (IL)-2, IL-4, IL-5, and IL-10. In total, 50 microliters of each sample were incubated for 2 h at room temperature with equal amount of capture beads and PE detection reagent. After incubation, the samples were washed once with washing buffer, and the pellets were resuspended in 200 µL washing buffer for analysis using a BD LSRFortessa Cell Analyzer. The concentration was analyzed within the range of the generated standard curve according to the manufacturer’s recommendations.

### 2.8. Determination of Antibody Titers by Hemagglutination Inhibition Assay

We performed the hemagglutination inhibition (HI) assay to measure influenza subtype-specific HI antibody titers. Receptor-destroying enzyme-pretreated serum was absorbed by Turkey RBCs before use. Next, 25 µL of two-fold serially diluted serum was incubated with 25 µL of subtype-specific influenza virus (4 HA units each) for 60 min at room temperature. A total of 50 µL of 0.5% Turkey RBC was added, and the mixture was incubated for 60 min at room temperature. The HI titer of each sample was determined as the highest dilution factor of the mixture in which hemagglutination was not observed; only the sample with HI titer ≥ 5 was included in the analysis because of the testing serial two-fold dilutions from 1:5 to 1:160.

### 2.9. Statistical Analyses

Immune cell counts were presented as the mean ± standard error of the mean (SEM) and HI titers as the geometric mean ± SEM. Statistical analyses were performed using GraphPad Prism version 8.0 software (GraphPad Software, La Jolla, CA, USA). A two-tailed Student’s *t*-test and Fisher’s exact test were performed to calculate the statistical significance for differences between groups. In all cases, a value of *p* ≤ 0.05 was considered statistically significant.

## 3. Results

### 3.1. Zingiberis Processum Rhizoma Enhances Flu Antigen-Specific Antibody Production in Cyclophosphamide-Induced Immunocompromised Mice

Cy-induced immunocompromised mice were orally administered distilled water or ZR (500 mg/kg) daily for 10 days, following vaccination with the 2017/2018 seasonal quadrivalent influenza vaccine. At 4 weeks after vaccination, the effect of ZR on the immune response to the influenza vaccine was assessed (Figure 2A). The flu-specific antibody subclasses of total IgG, IgG1, and IgG2a were compared between the groups. Influenza antigen-specific antibodies decreased in Cy-induced immunocompromised mice. However, the production of total IgG and IgG1 was significantly higher in the Cy + ZR group than in the Cy group (Figure 2B).

The HI assay was performed to assess the effect of ZR on the production of functional antibodies that inhibit influenza virus binding ability. Cy-induced immunocompromised mice showed lower HI titers against the four influenza strains than did the control mice (Figure 2C). ZR administration did not affect HI titers in Cy-induced immunocompromised mice (Figure 2C). However, the number of mice with HI-positive serum against H1N1 influenza virus significantly increased in the Cy + ZR group (Figure 2D). These results suggest that ZR could prevent H1N1 influenza infection in an immunocompromised state by enhancing antibody production.

### 3.2. Zingiberis Processum Rhizoma Restores T Helper Type 1/T Helper Type 2 Balance to a Normal State in Cy-Induced Immunocompromised Mice

To assess the effect of ZR on immune cell responses, splenocytes were stained and analyzed by flow cytometry at 4 weeks after vaccination. ZR did not change the composition of T cells, B cells, dendritic cells, and monocytes/macrophages between the Cy treatment groups. However, the proportions of neutrophils and eosinophils in Cy-induced immunocompromised mice receiving ZR administration were lower than those of the Cy group (Figure 3A).

To assess the effect of ZR on influenza antigen-specific Th1 and Th2 cytokine responses, supernatants obtained from influenza antigen-stimulated splenocytes were analyzed by CBA and compared between groups. The levels of TNF and IL-5 were significantly lower in the Cy + ZR group than in the Cy group, whereas the levels of other cytokines had not changed (Figure 3B). The Cy + ZR group showed an increase in the IFN-γ/IL-4 cytokine ratio, which indicates a shift in the immune response toward Th1 immunity and restoration to a normal state (Figure 3C). In addition, the IFN-γ/IL-4 ratio and influenza antigen-specific total IgG production were significantly correlated (Figure 3D). These results suggest that ZR could induce the Th1 immune response during the antibody production process, resulting in enhanced antibody production.

### 3.3. Zingiberis Processum Rhizoma Suppresses the Increase of Eosinophil in the Spleen before Vaccination after Cyclophosphamide Injection

To assess the effect of ZR on immune cell responses before vaccination, splenocytes were stained and analyzed by flow cytometry at the time of completion of ZR administration and after Cy injection. ZR administration decreased the eosinophil count and proportion before vaccination but other cells did not change compared with those in the Cy group (Figure 4).

## 4. Discussion

Most previous studies have focused on vaccination using adjuvants or different doses of antigens to increase antibody production in immunocompromised individuals. However, the underlying problem is the individuals’ weakened immunity against pathogens; therefore, we attempted to develop methods for restrengthening their immune systems. In this study, pretreatment with ZR extracts enhanced flu-specific antibody production in Cy-induced immunocompromised mice after flu vaccination and restored the influenza antigen-specific Th1/Th2 balance to the normal state. In addition, administration of ZR extracts suppressed eosinophil enrichment caused by Cy injection in the spleen.

The increase in eosinophils by Cy treatment has already been reported in clinical and animal experiments [23,24,25]. At the onset of immune responses to antigens, eosinophils can provide immunomodulatory cytokines to promote Th2 polarization, such as IL-4, IL-6, and IL-13 [26,27]. Therefore, the eosinophilia caused by Cy could induce an abnormal Th1/Th2 balance, which decreases antibody production after flu vaccination. Although further studies are needed, ZR might restore this abnormal flow through the inhibition of Cy-induced eosinophil enrichment.

Many bioactive compounds in ginger, such as gingerols, shogaols, and paradols, have been previously identified [28]. Intraperitoneal injection of 6-gingerols before an ovalbumin airway challenge suppressed the recruitment of eosinophils to the lungs in mice [16]. During the drying process, gingerols can be transformed into corresponding shogaols, which are the major compounds in dried ginger [28]. Moreover, 6-shogaols have an inhibitory effect on leukocyte infiltration into inflamed sites in mice [29]. Further, 10-shogaol inhibits angiotensin-converting enzyme, which are critical for viral entry in COVID-19 [30,31]. This result indicates that ginger intake may inhibit the pathogenesis of COVID-19. However, a study stating that dried ginger extracts could prevent systemic eosinophilia, not a local increase, has not been conducted yet. Dried ginger extracts could be applied to treat eosinophilia-associated Th2 immune diseases, such as atopic dermatitis and asthma.

Our study has two limitations. First, we were unable to verify whether the inhibition of eosinophilia affects the Th1/Th2 balance after vaccination. Second, we did not determine whether increased influenza-specific antibodies are effective in protecting against influenza infection.

To overcome these limitations, we plan to identify the effective compounds and underlying mechanisms that restore the abnormal increase in eosinophils and Th1/Th2 balance by Cy. This information could be used to combat the low effectiveness of vaccines against not only the influenza virus but also the coronavirus in immunocompromised individuals.

## Figures and Tables

**Figure 1 nutrients-14-01984-f001:**
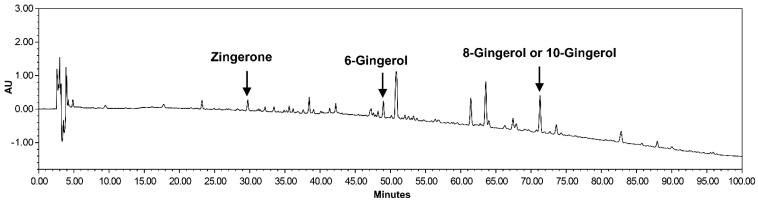
High-performance liquid chromatography of extracts from *Zingiberis Processum Rhizoma*. Identities of chromatogram peaks confirmed through their respective absorbance spectra obtained with the photodiode array 200 detector.

**Figure 2 nutrients-14-01984-f002:**
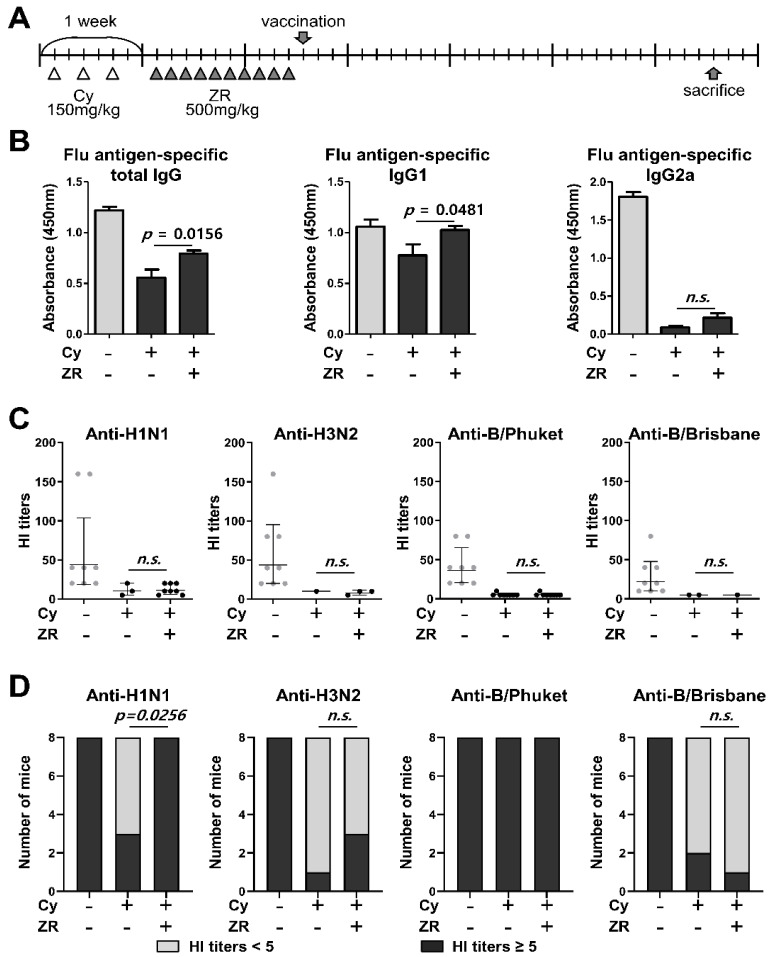
Evaluation of antibody responses to an influenza vaccination in ZR-administered immunosuppressed mice after flu vaccination. (**A**) Mice were intraperitoneally injected with Cy on days 1, 3, and 5 in the first week; mice injected with Cy (150 mg/kg) were then orally administered distilled water (*n* = 8, Cy group) or 500 mg/kg ZR (*n* = 8, Cy + ZR group) for 10 days, following vaccination with the 2017/2018 seasonal quadrivalent flu vaccine. The effect of ZR on antibody response after 4 weeks is shown. (**B**) Influenza-specific antibody titers (total IgG, IgG1, or IgG2) were measured using the enzyme-linked immunosorbent assay. (**C**,**D**) Influenza subtype-specific hemagglutination inhibition (HI) titers were measured using the HI assay. Non-treated mice were used as controls (*n* = 8, non-treated group). Cy, cyclophosphamide; ZR, *Zingiberis Processum Rhizoma*. Statistical analysis was performed using Student’s *t*-test and Fisher’s exact test.

**Figure 3 nutrients-14-01984-f003:**
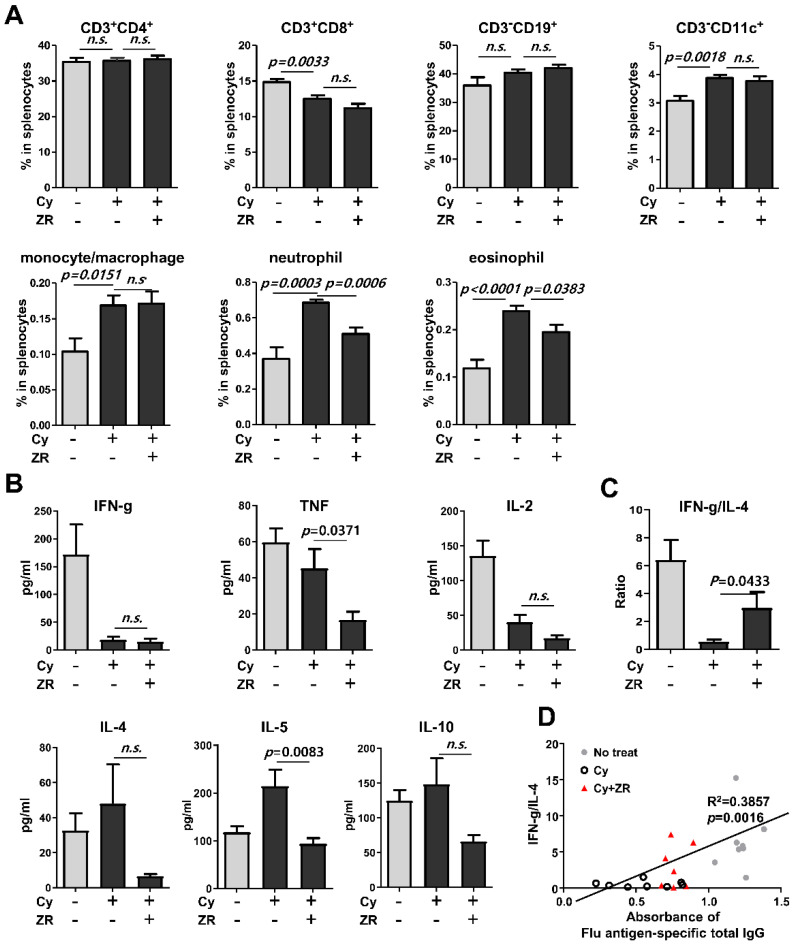
Analysis of immune cell responses in the splenocytes of ZR-administered immunosuppressed mice after flu vaccination. The effect of ZR on immune cell response assessed 4 weeks after vaccination (*n* = 8, each group). (**A**) Compositions of seven different types of immune cells were assessed in the isolated splenocytes from mice by flow cytometric analysis. (**B**) Antigen-specific cytokine levels were assessed by cytometric bead array in splenocyte supernatants after flu antigen stimulation. (**C**) T helper (Th) type 1/Th2 cell responses were assessed using interferon-gamma (IFN-γ)/interleukin (IL)-4 ratio in splenocyte supernatants after flu antigen stimulation. (**D**) The correlation between IFN-γ/IL-4 ratio and flu antigen-specific total IgG production. Cy, cyclophosphamide; ZR, *Zingiberis Processum Rhizoma*. Statistical analysis was performed using Student’s *t*-test.

**Figure 4 nutrients-14-01984-f004:**
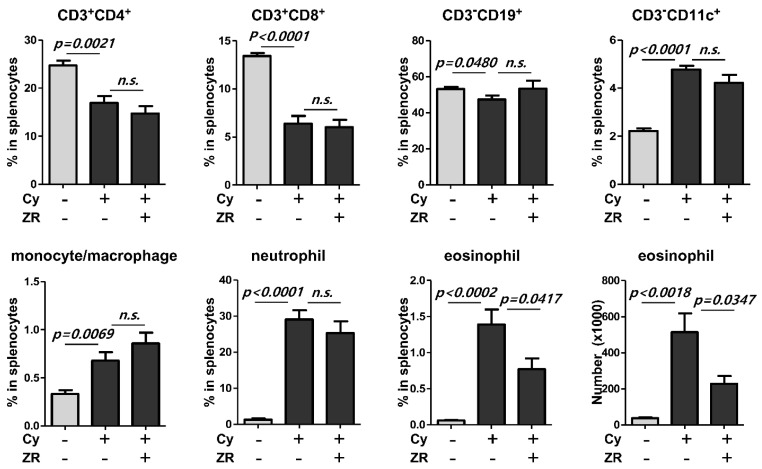
Analysis of immune cell composition in the splenocytes of ZR-administered immunosuppressed mice before flu vaccination. At the time of completion of ZR administration and after Cy injection, compositions of seven different types of immune cells were assessed in the isolated splenocytes from mice using flow cytometric analysis (*n* = 5, each group). Cy, cyclophosphamide; ZR, *Zingiberis Processum Rhizoma*. Statistical analysis was performed using Student’s *t*-test.

## Data Availability

The data presented in this study are available upon request from the corresponding author.

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
