# Peer review of "Dried Ginger Extract Restores the T Helper Type 1/T Helper Type 2 Balance and Antibody Production in Cyclophosphamide-Induced Immunocompromised Mice after Flu Vaccination"

_nutrients, 2022, doi:10.3390/nu14091984_

Round 1

Reviewer 1 Report

Comments to the Author

In my opinion, article requires general improvement. After corrections it may be reconsidered for publication.

  1. Latin italic names Zingiberis Processum Rhizoma.
  2. Write more about

Traditionally, ginger is udsed to prevent or treat the common cold and flu[8]. Ginger has been shown to have antiviral, anti-inflammatory, and antioxidant properties [9-12].

  1. Supplement the article with additional literature on this topic.
  2. Revise the literature as required by the journal.

Author Response

Reviewer #1

In my opinion, article requires general improvement. After corrections it may be reconsidered for publication.

  1. Latin italic names Zingiberis Processum Rhizoma.

à Thank you for pointing this out. We italicized Aingiberis Processum Rhizoma at all applicable places in the revised manuscript.

  1. Write more about

Traditionally, ginger is used to prevent or treat the common cold and flu[8]. Ginger has been shown to have antiviral, anti-inflammatory, and antioxidant properties [9-12].

à Thank you for your suggestion, Accordingly, we have added more text regarding this to the Introduction and Discussion section for easy understanding of the readers. We have added the findings that have been previously reported regarding the effects of ginger intake. (line 37-41, line 242-244)

“Traditionally, ginger is used to prevent or treat common cold and flu, but in recent times it has also been reported to exhibit therapeutic potential against COVID-19 through modulation of unbalanced T-cell responses. Ginger has been shown to have antiviral, anti-inflammatory, and antioxidant properties. It has been reported that its valuable functional ingredients like gingerols, shogaol, and paradols have anti-cancer potential.”

  • References
  • Jafarzadeh A. et al. Therapeutic potential of ginger against COVID-19: Is there enough evidence? Journal of Traditional Chinese Medical Sciences 2021 8(4):267-279.
  • De Lima R.M.T. et al. Protective and therapeutic potential of ginger (Zingiber officinale) extract and [6]-gingerol in cancer: A comprehensive review. Phytother Res 2018 32(10):1885-1907.
  • Mashhadi N.S. et al. Anti-Oxidative and Anti-Inflammatory Effects of Ginger in Health and Physical Activity: Review of Current Evidence. Int J Prev Med 2013 4:S36-42.

“Further, 10-shogaol inhibits angiotensin-converting enzyme, which are critical for viral entry in COVID-19 [30-31]. This result indicates that ginger intake may inhibit the pathogenesis of COVID-19..”

  • References
  • Afrida I.R. et al. Shogaol, Bisdemethoxycurcumin, and Curcuminoid: Potential Zingiber Compounds Against COVID-19. Biointerface Research in Applied Chemistry 2021 11(5):12869-12876
  • Haridas M. et al. Compounds of Citrus medica and Zingiber officinale for COVID-19 inhibition: in silico evidence for cues from Ayurveda. Futur J Pharm Sci 7(1):13.

  1. Supplement the article with additional literature on this topic.

à Thank you for your suggestion. We have added more text and their accompanying citations in the Introduction section for easy understanding of the readers. The added text highlights the previously reported results regarding the effects of ginger intake.

“Traditionally, ginger is used to prevent or treat common cold and flu, but in recent times it has also been reported to exhibit therapeutic potential against COVID-19 through modulation of unbalanced T-cell responses. Ginger has been shown to have antiviral, anti-inflammatory, and antioxidant properties. It has been reported that its valuable functional ingredients like gingerols, shogaol, and paradols have anti-cancer potential.”

  • References
  • Jafarzadeh A. et al. Therapeutic potential of ginger against COVID-19: Is there enough evidence? Journal of Traditional Chinese Medical Sciences 2021 8(4):267-279.
  • De Lima R.M.T. et al. Protective and therapeutic potential of ginger (Zingiber officinale) extract and [6]-gingerol in cancer: A comprehensive review. Phytother Res 2018 32(10):1885-1907.
  • Mashhadi N.S. et al. Anti-Oxidative and Anti-Inflammatory Effects of Ginger in Health and Physical Activity: Review of Current Evidence. Int J Prev Med 2013 4:S36-42.
  1. Revise the literature as required by the journal.

Thank you for your suggestion. We revised and checked the manuscript as per the journal requirements.

Reviewer 2 Report

The idea is interesting however there are a few drawbacks that make results and discussion questionable. The authors need to indicate how many mice were used? Why female mice only? How many were the experimental groups? Was treatment with ginger alone evaluated? Why was ginger extract administered only for 10 days? How long after i.p. Cy injections did the mice remain immunocompromised? How did the authors determine that mice were immunocompromised following Cy injections? Were mice injected only 3 times to induce immunosuppression? According to the time chart mice were injected 3 times during the first week of the experiment. Why was not thymus also studied?
